# *Ulva lactuca*, A Source of Troubles and Potential Riches

**DOI:** 10.3390/md17060357

**Published:** 2019-06-14

**Authors:** Herminia Dominguez, Erwann P. Loret

**Affiliations:** 1Department of Chemical Engineering, Faculty of Science, Campus Ourense, University of Vigo, As Lagoas, Galicia, 32004 Ourense, Spain; herminia@uvigo.es; 2Aix Marseille University, Avignon University Centre National de la Recherche Scientifique (CNRS), Institut de la Recherche et du Développement (IRD), Institut Méditerranéen de Biologie et d’Ecologie (IMBE), Unité Mixte de Recherche CNRS 7263 IRD 237 Faculté St Jérôme, Avenue Escadrille Normandie Niemen, Provence, CEDEX 20, 13397 Marseille, France

**Keywords:** *Ulva lactuca*, green tides, virus, bioactives, biofuels, biorefinery

## Abstract

*Ulva lactuca* is a green macro alga involved in devastating green tides observed worldwide. These green tides or blooms are a consequence of human activities. *Ulva* blooms occur mainly in shallow waters and the decomposition of this alga can produce dangerous vapors. *Ulva lactuca* is a species usually resembling lettuce, but genetic analyses demonstrated that other green algae with tubular phenotypes were *U. lactuca* clades although previously described as different species or even genera. The capacity for *U. lactuca* to adopt different phenotypes can be due to environment parameters, such as the degree of water salinity or symbiosis with bacteria. No efficient ways have been discovered to control these green tides, but the Mediterranean seas appear to be protected from blooms, which disappear rapidly in springtime. *Ulva* contains commercially valuable components, such as bioactive compounds, food or biofuel. The biomass due to this alga collected on beaches every year is beginning to be valorized to produce valuable compounds. This review describes different processes and strategies developed to extract these different valuable components.

## 1. Introduction

*Ulva lactuca* is a macroalgae and belongs to the phylum *Chlorophyta*, described by Linnaeus in the Baltic Sea in the seventeenth century [1]. It is able to grow attached, sessile or free floating. The capacity to reproduce involving two methods, one being sexual and the other being from fragmentation of the thallus, is rarely observed in macroalgae [2], but this provides a capacity for it to rapidly proliferate by covering the water surface, decreasing the biodiversity even for other algae species [3]. *Ulva lactuca* is a polymorphic species with morphologies dependent on the degree of water salinity [4] or symbiosis with bacteria [1].

Macroalgae (or seaweeds) contain high amounts of carbohydrates (up to 60%), medium/high amounts of proteins (10–47%) and low amounts of lipids (1–3%) with a variable content of mineral ash (7–38%) [5]. With decreasing available land and fresh-water resources, the oceans have become attractive alternatives for the production of valuable biomass, comparable to terrestrial crops. Macroalgae cultivated under controlled and sustainable cultivation systems are probably the future method for supplying biomass to meet market development needs [6].

The high carbohydrate fraction includes a large variety of easily-soluble polysaccharides, such as laminarin, alginate, mannitol or fucoidan in brown types; starch, mannans and sulphated galactans in red types and Ulvan in green types [7]. Alginate, one of the main structural polymers of brown seaweeds, provides both stability and flexibility for the specimens exposed to flowing water, and is one of the industrially-relevant carbohydrate compounds found in seaweed biomass, as are other hydrocolloids, such as agar and carrageenans, which are commonly used as thickeners, gelling agents or emulsifiers. Various other non-carbohydrate products obtained from seaweeds include protein, lipids, phenols and terpenoids, and minerals such as iodine, potash and phosphorus—useful for human and animal nutrition [8]. The harvesting of macroalgae—a valuable raw material for food—before they beach could well be developed into an effective solution [9]. The interest of macroalgae in human nutrition is due to their high mineral concentrations (such as calcium, magnesium and potassium) and glutamic acid, which makes them also useful as taste enhancers. Algae could help to address one of the biggest challenges currently faced by the food industry, which is the ever growing human population. Algae are also a source of active principles largely unexplored for pharmaceutical products [7]. The gelling properties of polysaccharides are well known, and the therapeutic applications are in development [10,11]. Algal polysaccharides, pigments, proteins, amino acids and phenolic compounds are potential functional food ingredients for health maintenance and the prevention of chronic diseases, with increasing potential uses in pharmaceutical industries [12].

*Ulva lactuca* polymorphism dependent upon the environment led to the proposal that different species may exist, such as *Ulva armoricana, rigida, prolifera, pertusa, fasciata* or *rotundata* [1,2,3,4]. However, genetic analysis revealed that the different phenotypes observed were not based on genetic variations that would justify the existence of different species other than *Ulva lactuca* [13]. Taxonomy is challenging for the genus *Ulva*, which belongs to the phylum of *Chlorophyta*, constituting four traditional classes (*Ulvophyceae*, *Trebouxiophyceae*, *Chlorophyceae* and *Chlorodendrophyceae*) that evolved from unicellular marine planktonic prasinophyte algae in the Neoproterozoic [1]. *Trebouxiophyceae* and *Chlorophyceae* are found mainly in fresh water while *Ulvophyceae* colonize mainly shallow marine environments, similar to the Australian *Collerpa* that has colonized the Mediterranean Sea since 20 years ago. Linnaeus was the first to observe that *Ulva lactuca* could have different phenotypes with a tubular or a sheet-like tail, but taxonomists in the nineteenth century proposed that the tubular green algae were a distinct genus called *Enteromorpha*. However, molecular approaches demonstrated that Linnaeus was correct—*Ulva* and *Enteromorpha* are not distinct genera [13]. Different *Ulva* species are still described in the literature [14], but it turns out that reproduction is possible between these different “species”, which therefore should be described as *Ulva lactuca* variants or clades.

*Ulva* blooms damage marine ecosystems and impair local tourism [9]. *Ulva* principally invades beaches, and its biodegradation can produce acidic vapors that have induced the death of animals and possibly humans since a horse was reported dead in 2009 on the Brittany coasts (located at the west of France) due to *Ulva* biodegradation [2]. The first *Ulva* bloom to be described was in Belfast (in the north of Ireland) at the end of the nineteenth century [15]. *Ulva* blooms were well-studied in the Laguna of Venice from 1930, with an unexplained decrease observed after 1990 [16,17]. Since 1980, *Ulva* blooms have been observed worldwide, from Galicia [18] to Tokyo Bay [19], including the American [20] and Australian continents [14]. However the largest events in the world to date have been the green tides observed in the Yellow Sea for ten consecutive years from 2007 and covering 10% of the Yellow Sea [21]. In Europe, Brittany’s north coasts have the biggest *Ulva* blooms [3]. While there is no doubt that *Ulva* blooms are due to human activities, it is generally farmers who are accused of being fully responsible for *Ulva* blooms because of their use of fertilizers [2,3]. However, for the green tides of Belfast and Venice, a correlation was established with human waste, due to an increase of workers [15] or tourists [17]. Furthermore, abundant sources of nitrogen and phosphorus are important for *Ulva* blooms, but phosphorus does not come from agricultural activity [2] and it is difficult to determine what part of nitrogen is due to fertilizers or human waste.

Preliminary experiments suggest that in the Mediterranean seas there is a virus that might provide a natural and ecological way to control *Ulva* blooms. *Ulva* blooms occur in the Mediterranean seas but disappear rapidly, and an enzyme activity related to giant viruses was observed in the bay of Marseille on denatured *Ulva* tissues (manuscript in preparation). Viruses are well known to participate in the control of microalgae blooms, but this is not demonstrated for macroalgae. Virus controls of microalgae blooms were recently observed in USA, with the two macroalgae *Aureococcus anophagefferens* inducing harmful bloom algae on the east coast [22] or *Tetraselmis* in Hawaii [23]. In these two cases, it was because of viruses that have been recently discovered and have been called giant viruses. Giant viruses were first discovered in amoebae [24]. While most viruses known over the past century have a size < 200 nm, such as 160 nm for HIV or 20 nm for the smallest virus (*Parvoviridae* infecting pigs), giant viruses have a size of up to 1 µm. Since then, giant viruses have been discovered all across the world, infecting many species, particularly marine species [25].

*Ulva* contains commercially valuable components susceptible of being exploited for cosmetic, pharmaceutical, chemical, food and energy applications [7,8,9,10,11,12,13,14,15,16,17,18,19,20,21,22,23,24,25,26] and this review provides a background on products that might be obtained from *Ulva*, as well as the processing technologies used to date.

## 2. Potential Riches of *Ulva*

The integral valorization could represent a more sustainable approach—contributing to biorefinery processes—to valorize different constituents of *Ulva* offering alternatives to global environmental concerns. Two aspects favor this utilization, valuable products found in *Ulva* and its high productivity [7].

### 2.1. Ulva Bioactive Compounds

*Ulva* protein hydrolysates show antioxidant [27], angiotensin-converting enzyme (ACE)-inhibitory activity [28] and immunomodulatory effects [29]. *Ulva* has starch (1–4%) as a reserve polysaccharide. Furthermore, *Ulva* has water-soluble and insoluble cellulose (38–52%) corresponding to structural polysaccharides with a major component called Ulvan. Ulvans are sulphated heteropolysaccharides that contribute to the strength of the cell wall and give flexibility to *Ulva*, preventing *Ulva* desiccation for tides. Ulvan is up to 30% *Ulva* dry weight and contains mainly sulfonic acids, sulphated l-rhamnose, xylose and glucose. This polysaccharide and its oligosaccharides have anti-viral, anti-tumour, anti-coagulant, anti-lipidique, hepato protective, immuno-stimulating, anti-depressant and anti-anxiolytic activities [26,30], and are increasingly requested for pharmaceutical and food applications. In addition, Ulvans are thermos reversible gels, with industrial applications in the chemical, pharmaceutical, biomedical and agricultural areas [26,27,28,29,30]. The commercial application of these gels is not yet as developed as other algal hydrocolloids [31]. However, interesting recent applications were reported with a composite material made of zinc oxide and calcium carbonate capped with *Ulva* polysaccharides for burn treatments [32]. *Ulva* carbohydrates can also be a carbon source for microbial production of biomaterials and building blocks to produce a range of chemicals and intermediates, such as organic acids, alcohols and biomaterials, but this market is still emerging [31].

*Ulva* has a unique fatty acid profile characterized by high levels of alpha linolenic acid (18:3ω-3) and stearidonic acid (18:4ω-3), the latter being an efficient precursor of eicosapentaenoic acid synthesis able to increase its human tissue levels [12,32]. In addition, *Ulva* contains phenolic, chlorophyll and carotenoids, which can be regarded as active free-radical scavengers [32]. The balanced content of Na and K (a ratio near to 1) is nutritionally beneficial [12]. A double-blinded versus placebo clinical trial showed that an *Ulva* water-soluble extract rich in oligo-elements (up to 50% and proteins of 10–20%) had a beneficial effect on depression [30].

### 2.2. Ulva for Food

*Ulva* can be a source of essential amino acids—some of them, such as histidine, are found in levels comparable to those found in legumes and eggs [12]. Traditional food uses the whole *Ulva*, which is also spread on agricultural land or composted, but these solutions rapidly reach their limits and have little added value. To eat *Ulva* from green tide is safe, and it is a valuable nutriment based on its high content of proteins and Fe, a good unsaturated lipid acid with a low-fat ratio and also has the presence of essential amino acids [21]. It is important, however, to verify that the concentrations of heavy metals, pesticides and polycyclic aromatic hydrocarbon is under regulation limits. *Ulva* from green tide could be used for a phytoremediation process of coastal water contaminated with bisphenol A [33]. The removal efficiency was positively influenced by light, nutrients and temperature, and salinity had no effect. This endocrine disruptor is rapidly and efficiently removed (more than 90%) by fresh *Ulva,* whereas less than 5% could be removed from dead algal biomass.

*Ulva* can be used to feed fish or mollusks such as abalone and reinforce immunity [34]. *Ulva* is now used in integrated multi-trophic aquaculture systems as a partial replacement or supplement for the diet of juvenile *Litopenaeus vannamei* [35]. Harvested *Ulva* contains a large amount of proteins (up to 30%) with a similar amino acid profile regarding commercial diet, but lower lipid levels (1.9%) and no docosahexaenoic acid. The replacement of up to 25% of the commercial diet with fresh *Ulva* does not impact shrimp production and juvenile shrimps fed with *Ulva* have a better growth rate, a higher carotenoid content and a lower lipid content, found in examining shrimp carcasses [36].

Agricultural utilization of *Ulva* extracts was reported to enhance the vegetative growth in bean plants under drought stress, limiting the lipid peroxidation, increasing the phenolic content and probably contributing to the enhancement of the antioxidant enzymatic activity [37]. *Ulva* aqueous extracts enhance the vegetative growth under drought stress conditions and antioxidant potential for *Salvia officinalis* [38].

The utilization of a single fraction from *Ulva* biomass has been explored and this alternative is recommended when the biomass is destined to a high value added application. Acidic *Ulva* extracts were used to replace synthetic antioxidants and to protect different cosmetic products from oxidation [39]. *Ulva* aqueous extracts were used for the synthesis of gold and silver nanoparticles, with excellent biocompatibility on healthy cells and were highly cytotoxic against colorectal cancer cell lines HT-29 and Caco-2 [40]. A new food-grade protein extraction protocol was proposed from *Ulva* and showed high digestibility in simulated gastro-intestinal tests with a high antioxidant activity [27].

Ulvan, proteins and amino acids offer key opportunities to deliver multiple products and energy through a biorefinery process [41,42]. In a study to explore the use of offshore grown macroalgae as a sustainable feedstock for biorefineries, it was estimated that the annual productivity for *Ulva* (838 g C/m^2^ year) was higher than the global average (290 g C/m^2^·year) estimated for terrestrial biomass in the Middle East [43]. Similarly, it was found that *Ulva* can offer the best performance for a future biorefinery (15 g dry weight/m^2^ d and 56 t/ha year) [44].

### 2.3. Production of Biofuels

The valorization of *Ulva* biomass for the production of biofuels is attracting attention regarding three aspects: bioremediation for the ecosystem, a renewable energy source and economic savings [45]. *Ulva* can be an attractive source for biofuels, since its production does not require arable land and fertilizers. *Ulva* can grow in saline and waste water and has a higher ability in sequestering atmospheric CO_2_ than terrestrial energy crops [46]. In addition, the growth rate and productivity are high compared to those of land crops, and they can withstand harsh conditions to survive under stressful conditions. The most studied biofuels are biodiesel, bioethanol and biogas. *Ulva* could be an alternative to conventional oil crops because they contain oil, suitable for esterification/transesterification reactions for the biodiesel production [47]. For biodiesel production, hexane is one of the most suited, but other organic solvents or their mixtures have been proven suitable for oil extraction from *Ulva* biomass [48].

After harvest, *Ulva* biomass requires pre-treatment and/or saccharification and fermentation to be converted in bioethanol. Carbohydrates contain hexose sugars, which are suitable materials for fermentation in producing ethanol. The hydrolysis of these polysaccharides results in monosaccharides, such as glucose, mannose and galactose. These carbohydrates are easily fermentable compounds in anaerobic digestion, and their extraction makes possible a rapid degradation. The quantities of cellulose and lignin, normally abundant in terrestrial biomass, are generally lower in *Ulva* and in the macro algal genera because of the different structural requirements in aquatic environments [49].

The production of the third-generation bioethanol from marine macroalgae depends mainly on the development of an eco-friendly and eco-feasible pre-treatment (i.e., hydrolysis), a highly effective saccharification step, and, finally, an efficient bioethanol fermentation step. *Ulva* has cellulose as a main structural component. Different hydrolysis processes are suitable to maximize the extraction of fermentable sugars, such as thermochemical hydrolysis with diluted acids (HCl and H_2_SO_4_) and a base (NaOH), and hydrothermal hydrolysis followed by saccharification with different fungal strains, preferentially adapted to the medium [50,51]. Also chemical-free treatments are beneficial. Efficient solubilization (over 90% of sugars) can be achieved by hot-water treatment, and further hydrolysis using cellulases and bioconversion is favored by the lack of enzyme and microbial inhibitors. In addition, nutrient supplementation is not required. Hot-water treatment of *Ulva*, followed by hydrolysis with cellulases, makes possible the production of fermentation media that can be easily converted into acetone, butanol and ethanol by a microorganism such as *Clostridium sp.* [52].

Pre-treatment (physical, chemical, enzymatic) of macroalgae has considerable influence on the technical, economic and environmental sustainability of biogas production. Different pre-treatments of *Ulva* for biogas production were compared, and the reducing sugar yields demonstrate that enzymatic pre-treatment was superior to acid catalysis, thermos alkaline and ultra-sonication [53]. *Ulva* wastes collected from coastal areas can produce up to 166 L CH_4_/kg volatile solids, whereas from food wastes and sewage sludge produce 350–380 L CH4/kg volatile solids [54]. Higher biogas yields from *Ulva* biorefinery facilities yielded up to 271 mL CH_4_/kg volatile solids (VS) in the range of methane production from cattle manure and crop land-based energies [41]. Enhanced bio methane production was observed in pretreated biomass and also in co-digestion processes [55].

A new promising alternative is the production of hydrogen from *Ulva* through dark or anaerobic fermentative technology [56,57]. The type of pre-treatment is important on the process performance, which can be improved with combinations of more than one pre-treatment and by the use of mixed anaerobic cultures [57].

## 3. Processes and Strategies to Extract Component from *Ulva*

Attempts to valorize *Ulva* collected on beaches leads to different problems, such as contamination with sands and pollutants from different origins. Industrial processes to produce artificially *Ulva* were developed to offer a production that is economically feasible and to attain a rational utilization of biomass following a biorefinery approach [58] since the sequential extraction of value-added products in a biorefinery is more efficient and viable [59]. The content of biologically active substances from natural origins depends on the site of the material collection, season and environmental conditions [60]. The medium nutrient content influences the fatty acid content; the ω-6/ω-3 fatty acids ratio and also the chlorophyll, carotenoid and phenolic contents, and therefore the antioxidant and anti-inflammatory properties [32]. Also, the extraction technology and operational conditions have a marked influence on the yields and on the properties of the target compounds.

In order to selectively separate the target components from natural materials, a solvent extraction stage is required. The solvent extraction process is a mass transfer unit operation from a solid material with a solvent, which shows preferential affinity for the target solutes. Different variables are important in the final yield: particle size, solvent type, solvent to solid ratio and temperature. These operational conditions require individual or combined optimization in order to maximize yields, the extraction rate and the purity and properties of the products.

### 3.1. Conventional Processes

Solvents can be different, depending on the polarity and location of *Ulva* components. An aqueous solvent is suitable for polysaccharides, which is the most abundant fraction. Oil and pigment fractions require less polar solvents. Ulvan is usually extracted with a high yield from hot water and pressurized conditions [61]. Ulvans are thermally stable until approximately 180 °C and present a high correlation between sulphate contents, their reducing power and their scavenging activity [62]. Water extraction of sulphated ulvan gives a product with a high antimicrobial activity against *Enterobacter cloace* and *Escherichia coli* [63]. A simple acidic method for extraction of Ulvan, with relatively low content of protein and high sulphate, is frequently used [30,31,32,33,34,35,36,37,38,39,40,41,42,43,44,45,46,47,48,49,50,51,52,53,54,55,56,57,58,59,60,61,62,63,64]. The properties and bioactivities of polysaccharides can be modulated by the extraction conditions; where rhamnose, glucuronic acid and glucose are the major monosaccharides obtained at 90 °C with 0.01 M HCl. Glucose is the major monomer at 150 °C with 0.1 M HCl, and the sulphate content is also influenced by temperature and acid concentration. These parameters have an effect on functional properties, such as oil holding capacity, foaming capacity, stability, antioxidant activity and pancreatic lipase inhibition activity, which are modified [65].

Ulvan extraction is better with a coagulation–sedimentation process to shorten the filtration time of the residual biomass after the extraction step, although this chemical treatment reduced the yield due to coprecipitation [66]. The extraction of proteins and glycoproteins from *Ulva* is more efficient with lysis solutions containing surfactants than with buffer or deionized water alone. The proteins further hydrolyzed with protease confirmed the availability and the lack of cytotoxicity in Vero cells [60]. The selection of the solvent system for oil extraction is an important factor for fuel production. Simultaneous distillation and extraction to prepare volatile compounds (aldehydes, ketones, carboxylic acids, alcohols and hydrocarbons) show antimicrobial activities and inhibition of tyrosine kinase [67].

An important group of bioactive compounds are phenols, although their content in green algae is lower than in brown types. They are usually extracted with organic solvents, the type of solvent being important on the extract yields and properties. Conventional solvent extraction with methanol or ethanol is used for the extraction of phenols and ethyl acetate [37,68,69,70,71]. Higher yields of pigment and phenolic were obtained using 95% ethanol, and carotenoids using acetone [32]. However, *Ulva* extracts show lower phenolic content and antioxidant properties compared with other natural materials or synthetic antioxidants [68].

### 3.2. Biorefineries

*Ulva* blooms represent a non-competitive green source for production of biofuels and other commodity materials. Abundant recent studies have confirmed the potential of *Ulva* for biorefinery. A general scheme following this strategy is shown in Figure 1. The major challenges for seaweed biorefineries are in relation with the production of high value products, the lower use of chemicals and waste disposal. The integrated biorefinery philosophy can solve different problems associated with the algal biomass conversion to bioenergy [46,72]. A marine biorefinery could be a solution to intensify *Ulva* production to obtain bioethanol [73]. The content of monosaccharides released by acid hydrolysis from different seaweeds was compared, revealing that *Ulva* has the highest economic potential [74].

Macro algal proteins and oligo- and polysaccharides are potential raw materials for the new generation of health ingredients having both techno- and bio-functional applications. Their extraction is usually placed in initial steps and the latter steps are usually those leading to biofuels [75]. The interest to extract valuable components present in the biomass in initial steps (phenolic and protein fractions) is to improve the economy of ethanol production and favor industrial implementation [76]. The utilization of the protein fractions would represent an opportunity for developing countries, which permanently face a protein shortage. The extraction processes should provide high protein yields, preserving the quality (amino acid profile and digestibility) and avoiding the presence of antinutritional compounds. Moreover, the extraction process should find its optimal placement in the whole bioethanol production chain.

Different washing steps have been proposed to remove salts for food applications [77]. After the evaporation of the washing water, the remaining solid biomass has a higher protein content from 11 to 24% and energy content from 20 to 50%. Regarding *Ulva* phenotypes, the content in salts goes from 29% to 36% with Na/K ratios from 1.1 to 2.2 and a maximum at 19% ulvan.

Mixing enzymatic and chemical extractions is proposed to maximize the extraction of high molecular ulvan fractions with gelling properties [78]. The adequate selection of the operating procedures (temperature and acid concentration) determines the chemical composition of ulvan and also the rheological and textural properties. *Ulva* insoluble dietary fiber shows high values in water holding capacity and oil holding capacity—comparable to other commercial fibers—which is suitable for the formulation of low caloric foods and in the stabilization of foods rich in fat and emulsion.

The feasibility was tested using *Ulva* residues remaining after the extraction of polysaccharides as an energetic source after a pre-treatment with hydrogen peroxide to enhance the efficiency of enzymatic hydrolysis and bioethanol yields [21]. The production of methane could be possible from the solid wastes of *Ulva* after sap extraction, Ulvan extraction and protein extraction, or after the sequential extraction of all these components. The highest methane yield of 408 L CH_4_/kg VS was observed in sap and Ulvan-removed residue, suggesting that the high protein and sulphate content are major inhibitors in anaerobic digestion [59].

An interesting valorization is obtained from leftovers due to the *Ulva* polysaccharide extraction process using hot water and enzymes [79]. This material was added to potato chip dough at a level of 2.5%. The water activity of chips with this *Ulva* extract was significantly less than other control chips without *Ulva*. The addition of *Ulva* leftovers increased the protein, ash and dietary fiber contents of baked potato chips. However, sensory scores showed that green algal addition produced an unacceptable color and strong seaweed flavor. The solid residue after the extraction of *Ulva* polysaccharides was also used to promote growth and to enhance non-specific immune and disease resistance enhancement against *Vibrio parahaemolyticus* in white shrimp *Litopenaeus vannamei* [80]. The animal mortality in the group fed with *Ulva* leftovers was lower compared with the control; the survival in the group fed with the polysaccharides extracted with cold water was 80%. In the group feed with the polysaccharides extracted with hot water, it was 65%, and in the control group, it was 40%.

An original biorefinery approach is proposed to isolate sugars from *Ulva* [81]. The sugars in the biomass are solubilized by hot water treatment followed by enzymatic hydrolysis and centrifugation, resulting in a sugar-rich hydrolysate, and a protein-enriched extracted fraction, which could be advantageously used in animal feed compared to intact seaweeds. The content in essential amino acids and digestibility suggest a promising use in diets for monogastric animals. Reduction of the high content in minerals and trace elements may be required to allow a high inclusion level of *Ulva* products in animal diets. The hydrolysate is used successfully for the production of acetone, butanol, ethanol and 1,2-propanediol [82].

A multistep integrated process for the extraction of different products from *Ulva* provides opportunities to develop novel products and commercial applications [83]. The protein content is 11% in dry weight and protein digestibility is 86%, indicating its suitability for use in food supplements. The cellulose extraction is the final step in this integrated approach and is the least affected by the up-stream treatments compared to other components. A similar strategy using *Ulva* leftovers after enzymatic saccharification proposes a use for aquaculture food [84]. During saccharification, the relative ash and carbohydrate content is reduced, but total nitrogen, total carbon and lipid content increase, making possible the survival and growth of bivalve spat and commercially valuable sea urchins over the course of three-week preliminary trials. Finally, another biorefinery approach integrates the hydrothermal liquefaction for biomass conversion to produce fuels (bio-oil and gas), aqueous fertilizers and remediation agents for domestic and marine culture effluents [85]. This technology is now used to transform macroalgae into aqueous phase products. *Ulva* offers the highest bio-oil yields. Hydrothermal liquefaction is effective for *Ulva* conversion, giving the highest bio-crude yields up to 29.9% and containing up to 60% of the total biomass energy content.

## 4. Emerging Technologies

The development of food grade, scalable, efficient and environmentally friendly extraction processes is required to reduce the energy and solvent consumption. It is important also to provide a sustainable utilization of the raw material, and to avoid the generation of toxic and undesirable compounds in the final products. Regarding these three purposes, the development of new techniques and the use of water as solvent are particularly promising [82,83]. The principles of these new techniques are described in Figure 2.

### 4.1. Pressurized Solvent Extraction

This technique is performed at a high temperature and pressure to maintain the solvent in liquid state faster, and it requires lower amounts of solvent than traditional extraction techniques. This technique is also named accelerated solvent extraction or high pressure liquid extraction. The use of food-grade solvents, such as ethanol, converts this process into a green technique suitable for an industrial scale, and is useful for food processing. The pressurized ethanol extraction of bio actives from *Ulva* shows reduced and antiradical properties and also shows growth inhibition of the food spoilage bacteria, such as *Escherichia coli*, *Micrococcus luteus* and *Brochothrix thermosphacta* [85]. The accelerated solvent extraction of proteins is performed after pre-removal of lipids with hexane and phenol with 70% acetone [83].

When the solvent used is water, the technique is also called superheated water, subcritical water extraction, high temperature water extraction, pressurized hot water extraction or hot water extraction. Water is used in a liquid state under pressure between 100 °C and the critical temperature of 374 °C. Temperature has an important effect on yield and selectivity. The dielectric constant of water decreases with temperature, enabling the extraction of compounds with different polarities at different temperatures. High temperature also enhances diffusivities, making possible the transport of solutes from the solid matrix, and a compromise solution with thermal degradation has to be reached. The major variable during the operation is temperature, at 150 °C or higher, since water properties (especially the polarity) are significantly affected and polymeric substrates can be hydrolyzed because of the enhanced reactivity of water under these conditions. This strategy is attractive for the conversion of aquatic biomass, since the energy-intensive drying steps could be avoided [86]. *Ulva* hydrothermal liquefaction shows substantial variation in composition, with a bio crude oil yield from 9 to 20% [85]. The hydrothermal liquefaction with subcritical water (300 °C) is very important to extract organic solvents, such as methanol and ethanol [52]. This technology is also suitable for treating a combination of microalgae and macroalgae for heavy metal bioremediation and bioenergy production [86]. The use of alcoholic solvents increased the bio-oil yield and converted the acids obtained in bio-oil to the corresponding methyl and ethyl esters. High-temperature liquefaction was used to improve the hydrolysis yield of *Ulva* sugars, which is already higher than other agricultural biomass [87]. This process generated less microbial inhibitors (HMF, 5-hydroxymethylfurfural) regarding other pre-treatments. After comparing different pre-treatments (organic solvents, alkaline solvents, liquid hot water and ionic liquids) to obtain an efficiency saccharification of *Ulva*, it appears that organic solvent and liquid hot water provide the highest glucan yield with 81% and 63%, respectively [88]. The most efficient saccharification before ethanol production is reached with the liquid hot water treatment with a 98% yield.

### 4.2. Microwave Assisted Hydrothermal Extraction

This technology is based on the use of electromagnetic radiation with frequencies in the range from 300 MHz to 300 GHz. This non-ionizing radiation generates heat by two mechanisms that are ionic conduction and dipole rotation, which usually occur simultaneously. Microwave is proposed for drying [89] and for the assisted hydrothermal extraction of sulphated polysaccharides. Microwave-assisted extraction using distilled water as solvent is used for the extraction of Ulvans [90]. This technique also offers operational and environmental advantages for the extraction of essential oils. Microwave-assisted hydro distillation was proposed to obtain essential oils with free-radical scavenging and antioxidant potential, as well as some components of medicinal importance to be used as a food additive and dietary supplement [91]. The transesterification of *Ulva* oils to biodiesel was obtained with significant yields in 1 h at 60 °C with sodium hydroxide and at room temperature with Na metal [92]. The microwave assisted transesterification could be completed within 5 min, but the highly exothermic and uncontrollable reaction of sodium metal in microwave damages the equipment. A microwave-assisted direct *Ulva* liquefaction using sulfuric acid was used as a catalyst to produce bio-oil [93]. Microwave-assisted liquefaction of *Ulva* under atmospheric pressure using a Fe_2_O_3_-modified catalyst increased the bio-oil yield and facilitated the production of long-chain compounds with a decrease in the oxygen content in the bio-fuel [94]. Microwave irradiation showed higher production of sugars compared with external conduction heating. Microwave irradiation efficacy for hydrolysis was also demonstrated with the addition of polyoxometalate clusters to produce larger amounts of reducing sugars than hydrochloric and sulfuric acids and inducing a lower amount (less than 0.1%) of furfural derivatives [95,96]. Phosphotungstic acid was found to be the most suitable agent and was recycled at least three times by diethyl ether extraction without changing its activity. Hydrothermal temperature until 140 °C was highly influential on the enzymatic *Ulva* hydrolysis and shortened the required reaction time [97].

### 4.3. Ultrasound

Ultrasound waves create compression and expansion when they go through a medium inducing cavitation. This phenomenon due to the production, growth and collapse of bubbles, which implode in the liquid near the solid surface of membranes, causes surface peeling, erosion and particle breakdown [98]. Intensification by ultrasound makes possible extraction due to a more effective mixing, and thermal and structural effects (fragmentation, erosion, sonocapillary, sonoporation, detexturation), as well as enhanced solute diffusion and washing. However, oxidation and degradation of compounds occurred as a result of solvent vapor, high pressures and temperature, which generate primary radicals susceptible of forming other radicals and molecules [99]. High power ultrasound can be an efficient tool for large scale applications because of improved equipment design and efficiencies of continuous operation. Ultra-sonication reduced the extraction time and solvent consumption, making possible a greater penetration of solvent into cellular materials and improving the release of cell contents into the bulk medium. In addition, this technique may also provide more economic and environmental as well as health and safety benefits [48]. The ultrasonic pre-treatment method improves extractions of oil with higher efficiency, reducing extraction time and increasing yields, with moderate costs and negligible toxicity. The ultrasonic pre-treatment method also reduces extraction time [48]. Solvent mixtures, i.e., diethyl ether and methylene chloride in n-hexane, can also favor high oil yields. Autoclave and ultrasound pre-treatments enhanced the oil extraction yield because they damage the *Ulva* membranes. Extraction of oil from *Ulva* biomass using the ultrasound pre-treatment method showed better results when compared with other methods, such as deep-freezing, lyophilization and microwave pre-treatment methods, which caused partial hydrolysis and pre-esterification of the oil. Using sonication in water (and subsequent ammonium sulphate-induced protein precipitation) provided the greatest protein yields, but the protein and fatty acids content was higher with the alkaline protein solubilization [87]. An optimized ultrasound-assisted extraction of crude *Ulva* sulphated polysaccharides showed radical scavenging, reducing power and a macrophage-stimulating capacity [100].

### 4.4. Supercritical Fluid Extraction

This technology is based on the utilization of solvents under conditions leading to the supercritical state, which presents physicochemical properties between those of the liquid and the gaseous state, facilitating the solubility and mass transfer. The preferred solvent is carbon dioxide, for its availability, cost, low toxicity and the possibility of using low temperatures. Carbon dioxide makes it possible to maintain thermos labile compounds and characteristics close to natural products. This has advantages, as the lack of solvent residues in the final product and the selective recovery of the target solutes in the supercritical extract would favor its potential use for food, pharmaceutical and cosmetic industries. Supercritical fluid extraction of *Ulva* was proposed to obtain carotenoids and phenolic compounds with antioxidant properties [101]. This technique provided a better quality for fatty acids, carotenoids and chlorophylls compared to conventional ethanol extraction. The presence of polyphenols due to this technique conferred antioxidant properties for cosmetic applications [102]. SC-CO_2_ extraction also provided efficient extraction of polyphenols [103]. Supercritical CO_2_ extraction made it possible to obtain extracts containing secondary metabolites with plant growth bio-stimulant properties. When applied by foliar feed on cress and seed maceration for wheat, the extracts enhanced chlorophyll and carotenoid content, root thickness and above-ground biomass.

### 4.5. Enzyme Extraction

This technology has improved since the last decade. Although the *Ulva* proteins and polysaccharides are usually extracted with water or polar organic solvents, the presence of complex polymers, accessibility problems and the attack of polysaccharides with hydrolytic enzymes damage the cell wall and release intracellular material. To improve enzyme extraction efficiency, new procedures have been developed, such as acid extraction and combined enzymatic-chemical extraction, providing antioxidant properties to ulvan [104]. Enzyme-assisted extraction was applied for *Ulva* bioconversion to produce antiviral and antioxidant extracts [105]. Endoproteases, as well as the combined mixture of carbohydrase, increased the extraction yields as well as the sugar and protein contents in the extracts, which showed free radical scavenging properties. The use of an enzyme-assisted extraction using commercial proteases and carbohydrases improves the extraction efficiency compared to water extraction at 50 °C [106]. This technique is less effective for polyphenol recovery, and increased from 0.41% to 0.99% in the dry weight (dw) hydrolysate, and it is promising for the extraction of Ulvans, which increased from 8.99% to 21.09% the dw hydrolysate. In addition, all hydrolysate fractions tested were non-toxic to human cells but were active against the *Herpes simplex* virus. Hydrolysis with cell wall enzymatic degradation offered different results depending on *Ulva* phenotypes [107]. The best results were obtained with the use of a polysaccharidase mixture (β-glucanase, hemicellulase, cellulose).

Ulvan is found in cell-walls, particularly between the two cell layers constituting the thallus, and ulvan depolymerization gives high-value oligosaccharides or fermentable monosaccharides for the production of bio-active compounds and bioethanol [108]. However, most tested enzymes are those commercially available, and further studies of Ulvan depolymerizing enzymes, such as ulvan lyases, are needed [109,110]. The extraction with water at 90 °C, sterilization at 121 °C, homogenization and enzymatic hydrolysis made it possible to obtain extracts with antiradical and reducing properties [111].

Pulsed electric field treatment is based on the application of a high voltage (up to 80 kV/cm) in short pulses at room temperature. This technique causes the formation of pores in the cell membranes of the biomass placed between two electrodes and is referred as electroporation or electro permeabilization. This phenomenon is due to the accumulation of opposite charges on cell membranes and their move under the influence of an electric field. Tissue disintegration caused by electroporation facilitates the release of intracellular compounds, enhancing the extraction yields such as with proteins [112]. Pulsed electric field treatment enhances dashing *Ulva* biomass using hydraulic pressing, providing a higher extraction of K, Mg, Na, P and S compared to pressing alone. High-voltage pulsed electric field treatment was more efficient for cell-membrane permeabilization and protein extraction than osmotic shock [113]. The protein concentration in the final product increased because of the removal of protein compounds, and the extract had a higher antioxidant capacity. However, incubation with pectinase and pulsed electric field treatment was less efficient for the extraction of water-soluble proteins and carbohydrates from *Ulva* compared to shear homogenization. The need for deep investigation of algal proteins and protein extraction technology for health is important, since osmotic shock combined with pulsed electric fields and hydraulic pressure produced extracts containing food allergens, such as superoxide dismutase, aldolase and thioredoxin [111]. Table 1 summarizes some examples of application of different technologies for the extraction of *Ulva* components.

## 5. Conclusions and Future Trends

This review shows that many different approaches have been developed to valorize *Ulva*. The potential valorization of *Ulva* to produce bioethanol was particularly studied. Ethanol is actually mainly commercialized as an adjuvant (up to 15%) to diesel (or gazole) fuel that does not require an adaptation of the diesel engine. It could be a tool to fight air pollution from vehicles because of its biodegradable nature, since ethanol contains 35% oxygen and adding oxygen to diesel results in more complete fuel combustion, reducing harmful tailpipe emissions [117]. Vehicles with an unleaded gasoline engine can be modified to use only ethanol, but this operation increases CO_2_ emission. The problem is that bioethanol can also be obtained from sugar cane or other agricultural productions at a much lower cost compared to *Ulva* biorefinery, regardless of the process. Furthermore, the use of diesel or other organic fuels should decrease to reduce CO_2_ emission. It is therefore probable that efforts to have a competitive source of bioethanol from *Ulva* could turn out to be useless in the future when electricity and hydrogen will replace organic fuels.

The valorization of *Ulva* wastes will also remain difficult for pharmaceutical drugs, cosmetics or food due to contaminants that have to be eliminated. New compounds useful for the pharmaceutical industry that could be isolated from *Ulva* will have to be chemically produced to be commercialized. It is important to note that screenings with biological tests should be preferentially carried out on the *Ulva* protein fraction, since it is easier to have authorization for a clinical trial with injectable synthetic proteins [118]. Furthermore, it could be possible to transform the biomass from *Ulva* with enzymes to obtain new products that may have interest for the food industry, as it possible now from vineyard biomass [119]. Plant glycosyltransferases are able to modify a range of small molecules, including terpenoids, phenolics and alkaloids found in *Ulva*. Finally, it would be possible to introduce into the *Ulva* genome genes from plants such as strawberry [119].

*Ulva* blooms will remain a source of troubles that could grow with increased global warming. However, there is a natural law called “kill the winner” that may interrupt this *Ulva* success story. When there is proliferation of a species, a predator of this species appears to control this proliferation. Among the most powerful natural predators, the biggest is not necessarily the most efficient. Viruses have demonstrated this, even for the human since the HIV has been found to be our most dangerous predator [120]. What is interesting with viruses is their specificity regarding their target, which is not the case for other predators such as humans, who can control or eradicate many different species. The apparition of a virus specific of *Ulva* may be a consequence of the high concentration of viruses in the Mediterranean seas. Viruses are the most abundant biological entities in sea waters that can be found even in the bathypelagic (1000 to 2000 m) zone, and the Mediterranean seas appear to have the highest concentration, mainly in the epipelagic (5 m) zone [121]. Prokaryotes and unicellular algae appear to be the main viral hosts, but only 9% of sequences obtained from the viral fraction had an identifiable viral origin, and no research has been carried out with sequences specific to giant viruses [121].

Finally, the use of bioinformatic tools based on a multi-omic modelling work could be useful to improve the *Ulva* valorization process, to predict *Ulva* blooms and to evaluate the ecological impact resulting from the introduction of a virus in a different ecological system. Compared to approaches applying machine learning on omic data directly, a multi-view approach merging experimentally and model-generated omic data can include key mechanistic information in an otherwise biology-agnostic machine learning process [122].

## Figures and Tables

**Figure 1 marinedrugs-17-00357-f001:**
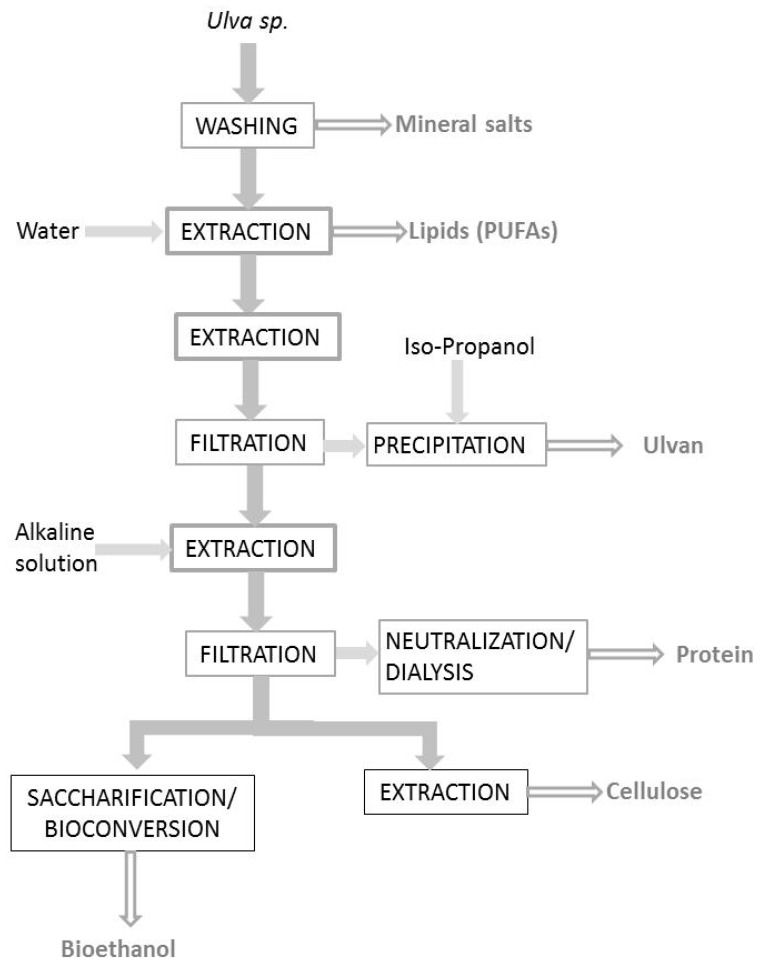
Simplified scheme of a biorefinery process according to final utilizations [83]. Different washing steps were proposed to remove salts. After evaporation of the washing water, the protein content increased by 11–24% and the energy content by 20–50%. The adequate selection of the operating procedures (temperature, filtration and chemical treatment) determined the chemical composition of extracts.

**Figure 2 marinedrugs-17-00357-f002:**
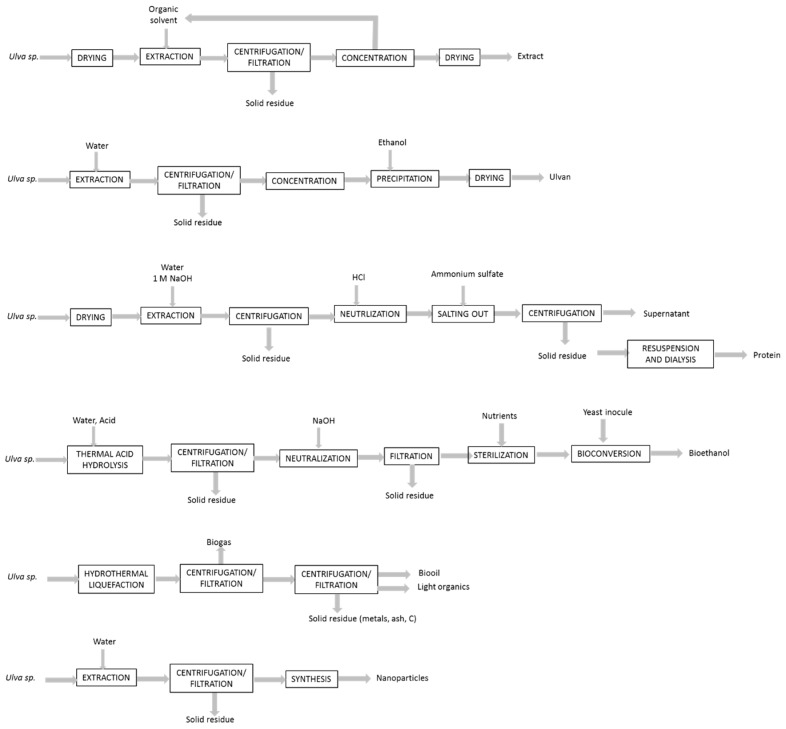
Examples of processes proposed to improve the extraction of *Ulva* components.

**Table 1 marinedrugs-17-00357-t001:** Examples of treatments for extraction and production of valuable components from *Ulva.*

Extraction and Pretreatment Stages of *Ulva lactuca (U. l.)*	Products	Ref.
*U. l. clathrata.* Extraction (0.2 N HCl, 60 °C, 2 h + 0.1 M NaOH, 60 °C, 2 h)	U, Pr	[64]
Supercritical (sc)-CO_2_ (50 MPa, 40 °C, 300–810 min)	A, Cy, M, Ph	[106]
*U. l. fasciata.* Ultrasound pretreatment (24 kHz, 50 °C, 6 min) + chloroform:methanol:hexane (90 °C, 3 h) or sc-CO_2_ (200 °C, 150 bar, 2.14 L/h)	O	[56]
Acid (1–3% H_2_SO_4_, 100–121 °C, 0.3–1 h) or alkaline (NaOH, 121 °C, 20.3 h) or buffer (pH 4.8, 120 °C, 1 h) or dry heat (120 °C, 1 h) or liquid ammonia (2%, 0.25 h) + enzyme saccharification + bioconversion	E	[114,115]
Dichloromethane, 7 days, 3 stages; adsorption, transesterification	BD	[97]
*U. l. flexuosa.* Sc-CO_2_ extraction (40 °C, 30 MPa, 11.4%)	Car, Chl	[104]
*U. l. intestinalis.* Liquefaction (deionized water, LSR 5; 345 °C; 30 K/min)	BO	[44]
US assisted extraction (66 °C, 40 min, LSR 50, pH 7.0)	U	[116]
*U. l.* Aqueous pretreatment (150 °C) + enzyme extraction + bioconversion	Pr, Ac, B, E	[81]
Water, thermal treatment (60 °C, 45 min) or water (pH 2, 95 °C, 3 h) or alkaline extraction (0.25M NaOH; 60 °C, 1 h) + anaerobic digestion	S, U, Pr, BG	[59]
Acid pretreatment (H_2_SO_4_, pH 2, 150 °C, 10 min); enzyme saccharification	Ac, B, E	[52]
MA water extraction (500 W, 70 °C, 20 min) or acid extraction (pH 1.5, 90 °C)	U	[62,95]
Bioconversion	BG	[54]
US water extraction or alkaline extraction or accelerated extraction	Pr	[87]
Incubation with water/heat treatment + lipid extraction + water extraction (90 °C, 2 h) + protein extraction (1 N NaOH, 80 °C)	S, O, U, Pr, C	[83]
US pretreatment (LSR 3, 24 kHz, 50 °C, 5 min); solvent extraction	O	[48]
OS (30 °C, 24 h) or enzyme (pH 4, 25 °C, 0.5% pectinase, 4 h) or PEF (7.5 kV/cm, 0.05 ms pulses, 6.6 kWh/kg prot)	Pr, U	[113]
Liquefaction (deionized water, LSR 5; 345 °C; 30 K/min)	BO	[44]
Water or buffer or PEF (75 pulses, 2.9 kV/cm, 5.7 μs, 0.5 Hz), pressing (5 min, 45 daN/cm^2^)	U, Pr	[111]
PEF (247 kJ/kg fw, 50 pulses, 50 kV, 70.3 mm electrode gap)	Pr	[115]
EA aqueous extraction (0.5%, 50 °C, 5 h)	Ph, Pr, U	[100]
*U. l. linza.* Extraction (pH 1.5–2, 60 °C, 2 h); re-extraction (pH = 2, 60 °C, 1 h)	U	[30]
*U. l. meridionalis* and *U. l. ohnoi*. MA hydrothermal extraction (4 + 10 min, 160 °C)	U	[94]
*U. pertusa.* Liquefaction (150 °C, 15 MPa, 15 min); enzyme saccharification	E	[91]
*U. l. prolifera.* MA liquefaction (6% H_2_SO_4_, 600 W, 30 min, 180 °C)	BO	[98]
MA liquefaction (ethylene glycol, 4.93% H_2_SO_4_), 600 W, 30 min, 165 °C	BO	[21]
*U. l. rigida.* Aqueous extraction (100 °C, 1 h)	Ph	[38]
Aqueous extraction (125 °C, 1.5 bar, 1 h) + SSF	U, Pr, E	[75]
EA extraction (pepsin + bromelain, 37 °C, pH 2.0, 20 + 20 h)	Pr	[28]
Acid hydrolysis (2–4% acid, 121 °C, 30–60 min)	E	[50,51]

EA: enzyme assisted; MA: microwave assisted; USA: ultrasound assisted; PEF: pulsed electric field; SSF: simmultaneous saccharification and fermentation; A: auxins; Ac: Acetone; B: butanol; BD: biodiesel; BG: biogas; BO: biocrude oil; C: cellulose; Car: carotenoids; Chl: chlorophylls; Cy: cytokinins; E: ethanol; M: minerals; O: oil; OS: osmotic shock; Ph: polyphenols; Pr: protein; S: sap; U: Ulvan.

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
