# Peer review of "Ulva lactuca, A Source of Troubles and Potential Riches"

_marinedrugs, 2019, doi:10.3390/md17060357_

Reviewer 1 Report

No comments anymore        

Author Response

Thank you very much

Reviewer 2 Report

Dear Authors,

in my opinion the manuscript reveals many "distractions" and only discusses a part of the recent bibliography about valorization of genus Ulva. The aim of the paper seems to be the use of Ulva biomass in biorefinery and for this reason informations on other topics are limited. The other aspects of the valorizations of biomass are reported with details that are not always sufficient for a Review. 

In the whole manuscript too often the scientific names are not in italics and this aspect deserves more attention.

The manuscript includes tables and figures that do not have a homogeneous format and which in my opinion they could be excluded or presented as Supplementary Materials.

In the bibliography it is necessary to modify several reports to correct the numerous errors and add some references.

The manuscript is not ready for publication but some suggestions are indicated in the attached file.

Author Response

Thank you very much and it was very kind to provide the text with the modifications.

Reviewer 3 Report

This review provides a background into the occurrence of Ulva lactucablooms, the range of products that might be obtained from them, and the processing technologies used to date.  I found the subject interesting, the range of techniques well covered with a good level of detail, and as such, the review is potentially useful to readers.  Although several reviews concerning Ulvahave been published in recent years, none of them take the same approach as the current authors, so the review is timely and useful. 

The organisation of the sections is reasonable; however the manuscript requires considerable work to improve the flow of the text (which meanders somewhat and needs tighter, more concise writing) and the written style is poor, including irrelevant or colloquial comments (lines 551-555), poor sentence construction and many spelling ("technic"), grammatical and formatting errors (e.g. lines 403-405, failure to subscript consistently, variable capitalisation of "Ulva" etc) which are not related to the fact that English may be a second language for the authors, and suggest that the manuscript was not carefully proofed before submission.

I started to note the errors for correction but there were simply too many of them.  The manuscript requires a thorough proof reading for spelling, grammar, style and format and will not be publishable until it is considerably more concise and free from errors.  Although some examples are pointed out below, simply correcting these will not redeem the manuscript.

The figures also need improvement.  Fig 1 is very basic and would benefit from expansion into several panels which illustrate, for example, photographs of Ulva lactuca specimens (particular with different morphologies since the authors discuss that issue in some detail), a photograph of an Ulvabloom, and/or a map of the major affected areas.  Also, the current Fig 1 is cited in the text as illustrating the occurrence of ulvan (Line 113) which it doesn't, as well as detailing some strategies for utilisation (line 174) which it barely does.  A figure should be of use and should provide more detail than the text which cites it.  Fig 2 is adequate but Fig 3 should provide the references which utilise each of the schemes outlined in the figure, so the reader can quickly identify the relevant papers.  Table 1 would benefit from left justification of the text and some clear delineation of the strategies pertaining to each reference because at present, it is hard to tell where the lines related to each section start and finish.

During the discussion of each technique, the review would be improved if specifics could be provided where possible (e.g. line 522 quantification might be possible for the term "less efficient").  Sometimes this is done well (lines 314-315) but other times the reader is left with qualitative statements (lines 337-338; 355-357; 410-411; 435-436; 458-461) where quantification might be possible. 

In line 95 the authors state that they have data suggesting that a virus may account for the control of Ulva lactucablooms in the Mediterranean. Given that they discuss this both in Lines 94-103 and also later in lines 550-563 it is unsatisfactory to justify this assertion by simply stating "data not shown". Even if the data (albeit preliminary) is not presented, they should indicate the nature of the data, and indicate whether it is being further investigated.

In lines 564-569 the potential of "multi-omic modelling" is touted to improve Ulvavalorization.  This paragraph is very general and does not explain how multi-omic data will be justifiable or will materially improve the processing of Ulvabiomass.  I would delete this paragraph unless some detailed expansion is going to be provided.  Similarly, in line 549 the suggestion that strawberry genes might be introduced to Ulva(thus producing GMOs) is rather outside the scope of the article (which proposes processing of Ulvabiomass as a solution to the problem of Ulvablooms). An article on Ulvagenomics and GMO construction is, in my opinion, a separate article.

In summary, the text needs a great deal of attention before it is publishable.  However, the subject and treatment are worthwhile.

Author Response

Dear Reviewer 3:

Thank you very much for your careful reviewing and your encouraging comments regarding our review on Ulva. I understand that you could consider the publication of this manuscript after a thorough proof reading for spelling, grammar, style and format. You will be amazed that I am pleased to see that you are not satisfied that no data are presented regarding that a virus may account for the control of Ulva blooms in the Mediterranean. You have to know that the first version of the review included preliminary data with microscopic views showing Ulva tissues becoming white after five days and statistical analyses showing that this phenotype was different regarding the location of the sea water sampling in the bay of Marseille. Moreover, among the three figures that I removed there was one similar to what you suggest with photographs of Ulva specimens changing its morphology from tubular to lettuce tale regarding water salinity. Two reviewers accepted the MS but asked me to remove these preliminary data and a third reviewer just rejected the MS (this reviewer was probably French due to Gallicisms in its comments…). We were encouraged to submit again this manuscript and I am pleased to see that this new version is satisfying the two former reviewers. Reviewer 2 kindly provided us a PDF version of the text detailing minor modifications she is asking for that covers most of your remarks.

The text was carefully proofed by a native English speaker (I thank him in the acknowledgment). Furthermore we added quantifications as you requested. We move forward in the isolation and the identification of this virus that is a Mimiviradea as I suspected (the manuscript is in preparation).

Best wishes,

Erwann Loret

Round  2

Author Response

Thank you again and I modified the MS as you requested. 

Reviewer 3 Report

Dr Loret, like you, I regret that the preliminary data concerning the virus have been removed from the first version of your review. However, it is acceptable to simply explain that a manuscript is in preparation.  I also regret the loss of the extra panels you describe in Fig 1 and I continue to support the need for more detail in Fig 1. However, I will leave it to the Editor to decide whether Fig 1 remains useful as it is. The additional quantifications you mention are welcome, and strengthen the manuscript.

The recommendations in my original review concerning the deletion or revision of the last two paragraphs ("kill the winner" and "multi-omics") still stand. If you prefer to leave these unchanged, please explain why you consider them to be important.

I have looked at the textual recommendations of Reviewer 2 and by and large I concur with them.  The matter of English style is always difficult when one is not a native speaker but nevertheless must be attended to. While the manuscript is definitely improved, I'm afraid your native English speaker (Dr McBitter I presume) has missed many problems with the technical writing which still remain. To clarify what I mean, I have provided an edited version of the first 2 pages as a pdf, but as I'm sure you'll understand, proof reading and grammatical correction is time consuming and is not the primary role of a journal reviewer, so further close attention to proof reading is needed to bring the manuscript to a publishable standard.

Author Response

Dear Reviewer 3,

Thank you for your help regarding my bad English. After I read your comments, I have to say that I lost my confidence in my capacity to write in English in spite of more than 60 accepted publications in peer reviewed scientific journals. Ian Mc Bitter says that my English is acceptable (certainly a polite way to say that he can understand my wharf English). He made a new proof reading and I hope that the modifications he made will make the MS suitable.

In the conclusion and perspectives, I wish to keep the sentences because I like the idea that mother Nature might find its own solution to resolve a problem that we induced. Some unknowns are still remaining such as to understand why Ulva blooms occur in some beaches and not in other in a same area and we hope that multi-omic modelling carried out by a British partner at Teesside University (he belongs to a European network that we constituted with Herminia and a German partner to study green tides and valorize seaweeds wastes) will be a great help to elucidate the different parameters that trigger Ulva blooms.

Best wishes,

Erwann Loret

This manuscript is a resubmission of an earlier submission. The following is a list of the peer review reports and author responses from that submission.

Round  1

Reviewer 1 Report

See atached

Reviewer 2 Report

The manuscript "Ulva lactuca, a source of troubles and protential riches"  has been critically reviewed with the following comments:

1) Suggestion would be to remove the part about macroalgae blooming (chapter/section 2) and focus on the biorefinery of macroalgae (chapters/sections 3, 4  and improve this part not only for biofuel but also for e.g. food, pharmaceutics, cosmetics, chemicals and the processing possibilities. The blooming part is a socalled contradiction as a virus might be the course of disapperance of this blooming and then it would not be handy to present in the same part the biorefinery of macroalgae components useful for all kind of applications. 

2) Put the blooming part of macroalgae in another manuscript and investigate more in detail including the disappearance of the macroalgae due to virus infections?

Reviewer 3 Report

Dear Authors,

I read your work and found it very confusing. You have chosen to submit your manuscript as Review but you also include some experimental data. Very unusual editorial form!

A Review must be a critical summary of the articles published on a specific topic but not report new data.

Therefore I believe that it would be better to work on two different projects, a Review in which you present and discuss the updated literature data regarding the valorization of Ulva, and another work focused on the proliferation of Ulva populations where to present your preliminary experimental data (still too preliminary) to which you add new ones.

As a result of this, I suggest rewriting your manuscript as a typical Review, modifying it entirely in light of the new goal.

Moreover:

- Many sections should be rewritten;

- Many references should be corrected according to editorial rules;

- Many scientific names must edited in italics.

As a result of this, I believe that the work presented must be rejected.